# Silica Shell Thickness-Dependent Fluorescence Properties of SiO_2_@Ag@SiO_2_@QDs Nanocomposites

**DOI:** 10.3390/ijms231710041

**Published:** 2022-09-02

**Authors:** Eunil Hahm, Ahla Jo, Sang Hun Lee, Homan Kang, Xuan-Hung Pham, Bong-Hyun Jun

**Affiliations:** 1Department of Bioscience and Biotechnology, Konkuk University, Seoul 05029, Korea; 2Department of Chemical and Biological Engineering, Hanbat National University, Deajeon 34158, Korea; 3Gordon Center for Medical Imaging, Department of Radiology, Massachusetts General Hospital, Harvard Medical School, Boston, MA 02114, USA

**Keywords:** fluorescence, silica shell, fine control, shell thickness, assembled structures, MEF

## Abstract

Silica shell coatings, which constitute important technology for nanoparticle (NP) developments, are utilized in many applications. The silica shell’s thickness greatly affects distance-dependent optical properties, such as metal-enhanced fluorescence (MEF) and fluorescence quenching in plasmonic nanocomposites. However, the precise control of silica-shell thicknesses has been mainly conducted on single metal NPs, and rarely on complex nanocomposites. In this study, silica shell-coated Ag nanoparticle-assembled silica nanoparticles (SiO_2_@Ag@SiO_2_), with finely controlled silica shell thicknesses (4 nm to 38 nm), were prepared, and quantum dots (QDs) were introduced onto SiO_2_@Ag@SiO_2_. The dominant effect between plasmonic quenching and MEF was defined depending on the thickness of the silica shell between Ag and QDs. When the distance between Ag NPs to QDs was less than ~10 nm, SiO_2_@Ag@SiO_2_@QDs showed weaker fluorescence intensities than SiO_2_@QD (without metal) due to the quenching effect. On the other hand, when the distance between Ag NPs to QDs was from 10 nm to 14 nm, the fluorescence intensity of SiO_2_@Ag@SiO_2_@QD was stronger than SiO_2_@QDs due to MEF. The results provide background knowledge for controlling the thickness of silica shells in metal-containing nanocomposites and facilitate the development of potential applications utilizing the optimal plasmonic phenomenon.

## 1. Introduction

Multifunctional nanomaterials, nanomaterials possessing multiple different properties in a single nanostructure platform, recently attracted great attention because of their potential applications in biology [1]. However, the structure and components of multifunctional nanomaterial must be carefully designed to maximize their intrinsic functions [2]. Among different functional nanomaterials, the integration of plasmonic metal nanoparticles (NPs) and fluorescent NPs in multifunctional nanomaterial is a promising combinatorial approach as they complement each other and overcome their intrinsic weaknesses [3]. For example, fluorescence bleaching can be overcome when plasmonic and fluorescence properties combine in multifunctional nanomaterials. The presence of plasmonic metal nanostructures such as gold and silver NPs can enhance fluorescent intensities via metal-enhanced fluorescence (MEF) or surface-enhanced fluorescence (SEF) phenomena [4]. As a result, many studies recently reported plasmonic-enhanced fluorescence properties for bioapplications [5,6,7,8,9,10,11,12,13,14,15,16,17,18,19,20,21].

According to the literature, fluorescent intensities increase dramatically between a metal surface and a fluorophore at a distance of 20 to 50 nm [22,23,24]. On the contrary, the quenching effect appears when the metal–fluorophore distance is below 20 nm because of the electron transfer from the excited fluorophore to the metal surface [22,23,24]. Therefore, it is necessary to develop appropriate synthesis methods, maintaining an optimum distance between the plasmonic and fluorescent components to maximize the fluorescent property of hybrid plasmonic-enhanced nanomaterials and avoid metal-induced quenching. In addition, the methods should be simple and cost-effective with options for various functionalizations. Various methods have been developed via tuning the separation distance between the fluorophore and the metallic surface in controlling plasmonic-enhancement effects [25,26,27]. For example, Liang et al. used charged polyelectrolytes to control the distance between the polymeric fluorophore and Ag nanocubes [25]. In addition, various silica (SiO_2_) shells (up to 30 nm) were coated on Ag NPs to investigate the interparticle distance-dependent fluorescence of Au nanoclusters [26]. Ke et al. reported the MEF and metal-enhanced singlet oxygen generation of SiO_2_-coated Au nanorod core–shell structures with aluminum phthalocyanine [28]. Among them, the core–shell structure provides spatial separations between plasmonic NPs and fluorophore for the MEF and also chemically stabilized metal NPs [26,29].

SiO_2_ is the most widely used material for shell coatings [30,31,32,33] due to its cost-effectiveness, chemical inertness, easy surface modification, biocompatibility, and optical transparency [34]. As a result, many studies on SiO_2_ coatings of metal NP structures have been reported. For example, Tian et al. prepared Au@SiO_2_ core–shells and etched its surface to create a pinhole structure [35,36]. Moreover, the presence of thin SiO_2_ shells prevents the adsorption of molecules on the Au core and simultaneously improves thermal stability [37,38,39].

However, the precise control for SiO_2_ shelling of nanocomposites remains a major challenge. In particular, the SiO_2_ shell’s thickness in nanocomposites, which include metallic components and fluorophores or Raman-labeling compounds, greatly affects their physicochemical properties such as plasmonic quenching, MEF, and SERS, which depend strongly on the distance between the metallic surfaces and the molecules with specific properties [24,40,41,42]. Nevertheless, studies on SiO_2_ shell’s thickness control and optical properties of NPs have been mainly conducted on single metal NPs [35,43,44,45]. Furthermore, only a few reports have been published on the fine-tunable control of the SiO_2_ shell thicknesses of metallic nanocomposite and their effects from the metal surface to fluorophores on fluorescence characteristics.

Ag NP-assembled SiO_2_ (SiO_2_@Ag) has been developed by our group as a basic platform for fluorescence, SERS, and magnetism [2,46,47]. The desired absorption wavelength of SiO_2_@Ag nanocomposite can be tuned efficiently by the presence of assembled Ag NPs on the SiO_2_ surface [48,49].

In this study, the number of NPs and the amount of SiO_2_ precursor were investigated to finely modulate the thickness of SiO_2_ shell on the surface of SiO_2_@Ag without structural destruction. In addition, quantum dots were introduced into a SiO_2_ shell-coated SiO_2_@Ag nanocomposite, and their optical properties according to the thickness of the shell are reported. This study presents a valuable approach that can provide optimal conditions for fluorescence enhancement through the fine control of the SiO_2_ shell thickness of nanocomposites. Our results show that single metal nanoparticles as well as nanocomposites exhibit the presence of MEF phenomena according to the SiO_2_‘s shell thickness.

## 2. Results and Discussion

SiO_2_@Ag as nanocomposites were synthesized as pre-reported [50,51]. Briefly, SiO_2_ NPs (153 ± 2.4 nm) were prepared by the modified Stöber method and were incubated with 3-mercaptopropyl trimethoxysilane (MPTS) to convert hydroxyl groups to thiol groups. Ag NPs were assembled on the SiO_2_-SH surface by reducing silver nitrate (AgNO_3_) in ethylene glycol (EG) with octylamine (OA). The transmission electron microscope (TEM) images of the SiO_2_ and SiO_2_@Ag NPs, shown in Appendix A, confirmed their uniform size and shape and excellent dispersion in ethanol without aggregation. The SiO_2_@Ag NPs exhibit their rough surfaces due to the assembly of Ag NPs. The average diameter of SiO_2_@Ag NPs, as measured by ImageJ software, is 188 ± 7.3 nm. The optical properties of the NPs were investigated by using UV-Vis spectrophotometry. Appendix A shows the UV-Vis absorbance spectra of the SiO_2_ and SiO_2_@Ag NPs. The UV-Vis absorbance of SiO_2_ NPs decreased rapidly from 300 nm to 1100 nm, corresponding to the absorbance of typical SiO_2_ NPs [52]. The UV-Vis absorbance of SiO_2_@Ag exhibits a broad localized surface plasma resonance (LSPR) band from 325 nm to 1100 nm and a maximum UV-Vis absorbance in the wavelength range from 400 to 500 nm, which corresponds to the absorption region of Ag NPs [53]. This result confirmed that Ag NPs were assembled on the SiO_2_ surface, and the nanocomposite SiO_2_@Ag could absorb light from the visible to near-infrared region. According to the literature, LSPR from visible light to ultraviolet regions represents highly sensitive wavelengths depending on NP’s component, shape, and ambient mediums [54]. Therefore, the optical property of SiO_2_@Ag can be controlled by experimental conditions in our research. The thickness of the SiO_2_ shell is highly sensitive to experimental conditions. Therefore, it is difficult to evenly form a perfect silica shell with sub-nanometer thickness on the NPs because excessive reactions for the formation of non-core silica nanostructure such as silica NPs must be excluded. Therefore the SiO_2_ coating on the SiO_2_@Ag surface was controlled and applied by two experimental parameters, the amount of SiO_2_@Ag NPs itself or the amount of SiO_2_@Ag NPs combined with the amount of SiO_2_ precursor, according to a previously reported method [55]. Sodium silicate (Na_2_SiO_3_) and tetraethyl orthosilicate (TEOS) were used as SiO_2_ precursors. The morphology and structure of SiO_2_@Au@SiO_2_ with different silica shell thicknesses are observed by TEM images (Figure 1A, Figure 1 and Appendix A). The observed silica shell layer on SiO_2_@Ag surface is quite homogeneous and ranges from 4 nm to 38 nm.

First, the formation of a silica shell from Na_2_SiO_3_ through solvent exchanges is a result of a sharp decrease in the solubility of Na_2_SiO_3_ in a mixture of water and ethanol, and controlling this process is difficult. Therefore, a small change of Na_2_SiO_3_ can also lead to a significant change in the formation of the silica-shell layer. Therefore, we fixed the amount of Na_2_SiO_3_ and changed the amount of SiO_2_@Ag to prevent the formation of non-core silica nanostructure, such as silica NPs. The amount of SiO_2_@Ag NPs was adjusted by changing the quantity of SiO_2_ from 5 mg (Figure 1A and Appendix A), 10 mg (Figure 1A and Appendix A), and 20 mg (Figure 1A and Appendix A), while 14.4 µL Na_2_SiO_3_ was fixed. By controlling the amount of SiO_2_@Ag NPs, thin silica shells were obtained in the range of 4 to 13 nm (Figure 1A and Appendix A: 4 ± 0.3 nm, Figure 1A and Appendix A: 9 ± 1.1 nm, Figure 1A and Appendix A: 13 ± 1.2 nm) without a leakage of Ag NPs from their surfaces. When the amount of SiO_2_@Ag decreases, the silica shell becomes is thicker. However, a decrease in SiO_2_@Ag NPs amounts leads to a low yield of SiO_2_@Ag@SiO_2_ products.

To generate a thicker silica shell on the SiO_2_@Ag surface with an expected product yield, TEOS was added into the S2 suspension to generate the samples Figure 1A, S4, S5, and S6. The formation of silica shell from TEOS occurs slowly and controllably through sol–gel processes. Therefore, we fixed the amount of SiO_2_@Ag and changed the amount of TEOS. Smooth and thicker silica shells with variable thickness were grown on SiO_2_@Ag NPs by the addition of different concentrations of TEOS at 1.2 mM (Figure 1A S4), 2.3 mM (Figure 1A S5), and 4.7 mM (Figure 1A S6). The thicknesses of silica shells on the SiO_2_@Ag@SiO_2_ are 16 ± 1.0 nm (Figure 1A S4), 24 ± 1.3 nm (Figure 1A S5), and 38 nm ± 2.0 nm (Figure 1A S6). These results demonstrated that the increase in TEOS concentrations leads thicker silica shells on the SiO_2_@Ag surface (Figure 1B). Moreover, the typical UV-Vis absorbance spectrum of SiO_2_@Ag@SiO_2_ was observed in Appendix A. Similarly to SiO_2_@Ag, the UV-Vis spectra of SiO_2_@Ag@SiO_2_ also broaden from 325 to 1100 nm but its absorbance intensity slightly decreased.

The study on the effect of silica shell thickness on fluorescence properties was performed by introducing 7 mg of quantum dots (CdSe@ZnS, QDs, QY 96.2%) on the surface of SiO_2_@Ag@SiO_2_ possessing different silica-shell thicknesses to generate various quantum-dot-assembled SiO_2_@Ag@SiO_2_ (SiO_2_@Ag@SiO_2_@QDs) (Figure 1B). For the introduction of QDs, the surface of SiO_2_@Ag@SiO_2_ was modified to a thiol group by MPTS. The morphology and size of SiO_2_@Ag@SiO_2_@QDs was confirmed by TEM images (Figure 2). As the silica shell became thinner, the introduced QDs were agglomerated and had a rough shape. In contrast, the aggregation decreased and distributed evenly as the silica shell was thicker. Moreover, the size of SiO_2_@Ag@SiO_2_@QDs increased with the introduction of QDs (QS1: ca.240.3, QS2: ca.252.2, QS3: ca.265.4, QS4: ca.270.1, QS5: ca.288.5, QS6: ca.306.8). These results show that QDs can be introduced into SiO_2_@Ag@SiO_2_ surfaces to generate SiO_2_@Ag@SiO_2_@QDs. When QDs were assembled on the surface of SiO_2_@Ag@SiO_2_, the maximum emission wavelength of QDs was blue shifted from 620 to 617 nm.

Figure 3 shows the change of fluorescence intensity by metals according to shell thicknesses. The emission spectra in Figure 3 showed that the fluorescent intensities at 620 nm of QDs significantly increased with silica-shell thicknesses and reached a maximum value at the 14 nm silica shell. Fluorescence intensities tended to decrease as shell thicknesses exceeded 14 nm. To compare the effect of metals on the fluorescence of QDs, SiO_2_ NPs with similar sizes were introduced by QDs (Appendix A). When the distance between the QDs to Ag NPs was less than ~9 nm, the fluorescence intensities of SiO_2_@Ag@SiO_2_@QDs were weaker than those of SiO_2_@QDs. However, it was observed that the fluorescence intensities of SiO_2_@Ag@SiO_2_@QDs were stronger when the distance between QDs and Ag NPs was between about 9 nm and 14 nm, and then it weakened again as this distance increased. The increase in fluorescence intensity with the silica shell thickness can be explained by two effects that affect the emission spectrum, which varies with distance [56]: the emission quenching [57] of metal NPs for the photoexcited QDs due to resonant energy transfers and the fluorescence enhancement of NPs promoted by the excitation of their localized surface plasmon resonances [58]. These phenomena, quenching and MEF, occur when the distance between the metal NPs to the fluorophores is within 20 nm [59], but details about the distance have not been revealed. It is important to note that small changes between metal NPs and fluorophores particularly affect metal emission properties. The only variable influencing is the plasmonic electromagnetic field decay exponential with the distance to the metal surface. Quenching is a phenomenon in which Förster resonance energy transfer (FRET) occurs when the distance between the metal NPs and the fluorophore is less than 10 nm. Therefore, it is considered that the fluorescence intensity of SiO_2_@Ag@SiO_2_@QD is weaker than that of SiO_2_@QD due to the quenching process caused by a distance between Ag NPs and QDs that is less than 10 nm [60]. Moreover, it is believed that the resonance excitation of LSPR on Ag NPs on the SiO_2_ surface generates an enhanced local field when the distance between Ag NPs and QDs is 9–14 nm, which greatly increases the fluorescence intensity of QDs. Therefore, when the distance between the QDs and the Ag NPs increases within 14 nm, it is estimated that the fluorescence signal is strengthened due to the increase in the MEF and a decrease in the quenching effect.

## 3. Materials and Methods

### 3.1. Materials

Ethyl alcohol (EtOH, 99.5% and 95%), tetraethyl orthosilicate (TEOS), ethylene glycol (EG), 3-mercaptopropyl trimethoxysilane (MPTS), silver nitrate (AgNO_3_), polyvinylpyrrolidone (PVP, average molecular weight ≈ 40,000), and octylamine (OA) were used without further purification. Ammonium hydroxide (NH_4_OH, 25~28%) was purchased from Daejung Chemicals & Metals Co., Ltd. (Siheung, Korea). CdSe@ZnSs (QDs) were obtained from Zeus (Osan, South Korea), and 18.2 Ω water was obtained using a Direct-Q Millipore purification system (SAM WOO S&T Co., Ltd., Seoul, Korea).

### 3.2. Methods

#### 3.2.1. Preparation of Ag-Assembled Silica NPs (SiO_2_@Ag)

SiO_2_ NPs (150 nm) were prepared using the modified Stöber method [61]. The surface of SiO_2_ NPs was converted to thiol groups by incubating SiO_2_ NPs (4 mL, 50 mg·mL^−1^ suspension in EtOH), MPTS (200 μL) and NH_4_OH solutions (27%, 40 µL). The suspension was stirred at 700 rpm at 25 °C for 12 h. Next, the suspension was centrifuged at 8500 rpm and washed several times with EtOH. The thiolated SiO_2_ NPs (SiO_2_-SH) was dispersed in EtOH and the final concentration was adjusted to 50 mg·mL^−1^. Ag NPs were added on the surface of the thiolated SiO_2_ NPs by a reduction of AgNO3 in the presence of PVP. The SiO_2_-SH NPs (0.6 mL, 50 mg·mL^−1^ in EtOH) were added in an EG solution containing PVP (5 mg), AgNO_3_ (26 mg), and octylamine (41.4 µL). The suspension was stirred at 700 rpm at 25 °C for 1 h. Then, the suspension was centrifuged at 8500 rpm and washed several times with EtOH. SiO_2_@Ag was dispersed in EtOH.

#### 3.2.2. Preparation of Silica-Shell-Coated SiO_2_@Ag with Various Thicknesses (SiO_2_@Ag@SiO_2_)

Various amounts of the SiO_2_@Ag (5, 10, and 20 mg) were separately dispersed in EtOH (1 mL) to prepare samples S1, S2, and S3. Distilled water (15 mL) containing 14.4 μL Na_2_SiO_3_ was mixed with the above SiO_2_@Ag suspension. The prepared suspension was stirred at 700 rpm for 1 h. A 60 mL aliquot of EtOH was then added to the resulting suspensions for solvent exchange. After 3h, parts of sample S2 containing 10 mg of SiO_2_@Ag were mixed with different concentrations of TEOS (1.2 mM, 2.3 mM, and 4.7 mM) under stirring at RT for 24 h to prepare samples S4, S5, and S6. After stirring, the above suspension was centrifuged at 8500 rpm and washed with EtOH to remove excess reagents.

#### 3.2.3. Introduction of QDs onto the Surface of SiO_2_@Ag@SiO_2_ (SiO_2_@Ag@SiO_2_@QD)

The surfaces of SiO_2_@Ag@SiO_2_ with various shell thickness (1 mL, 10 mg·mL^−1^) in EtOH were converted to thiol groups by vortexing MPTS (50 μL) and NH_4_OH (1%, 50 μL) for 1 h at 50°C to prepare thiol-modified SiO_2_@Ag@SiO_2_. The QDs (7mg), thiol-modified SiO_2_@Ag@SiO_2_ (10 mg), and a DCM (4 mL) were injected to a vial in series, and the mixture was vigorously vortexed for few seconds and shaken for 3 h. The resulting mixture was washed several times with EtOH and re-dispersed in EtOH (2 mg·mL^−1^) to obtain SiO_2_@Ag@SiO_2_@QDs.

#### 3.2.4. Physical Property Analysis of NPs

The size and morphology of prepared NPs were measured by transmission electron microscope (Libra 120, Carl Zeiss, Jena, Germany). The prepared NPs were well dispersed in EtOH (1 mg·mL^−1^). Then, 10 μL of the sample was dropped and dried at 25 °C on a 400-mesh copper grid (Pelco, Presno, CA, USA). The thicknesses of the SiO_2_ shell were analyzed by digitalized measurements using Image J software (v.1.53k, Bethesda, MD, USA). The average size of the NPs and the thickness of the SiO_2_ shell were calculated after analyzing at least 50 NPs.

#### 3.2.5. Measurement of UV-Vis Absorption Spectra

The particles were well dispersed in EtOH to obtain a suspension of 2 mg·mL^−1^ nanoparticle and transferred to a cuvette. UV-Vis absorption of the sample was performed in the wavelength from 300 to 1100 nm at the scanning speed of 1 or 5 nm/s by using a UV-Vis spectrophotometer (Mecasys OPTIZEN POP, Daejeon, Korea)

#### 3.2.6. Fluorescence Analysis of SiO_2_@Ag@SiO_2_@QDs

The fluorescence emission spectrum was analyzed by using a Cary Eclipse Fluorescence Spectrophotometer (Agilent Technologies, Santa Clara, CA, USA). An NP suspension (1 mg·mL^−1^) measuring 300 μL was added in a 96-well plate. The excitation wavelength was set at 385 nm. The sample was excited for 10 s and the fluorescence of the sample was collected in the range from 550 to 700 nm for 10 s.

## 4. Conclusions

A fascinating approach to silica-shell coatings on SiO_2_@Ag, one of the nanocomposites for multilayer synthesis, allowed us to control silica-shell thicknesses in a wide range from 4 nm to 38 nm. The thickness of the silica shell was finely controlled with two methods: controlling the amount of SiO_2_@Ag itself or combining the amount of SiO_2_@Ag and the silica precursor. Silica-shell thicknesses on the SiO_2_@Ag surface were measured to be 4 ± 0.3, 9 ± 1.1, 13 ± 1.2, 16 ± 1.0, 24 ± 1.3, and 38 ± 2.0 nm. As a result of testing, the effects of metal on the fluorescence according to the distance using the difference in the silica-shell thickness, the quenching effect, and MEF were observed with at a distance between Ag NPs to QDs within about 14 nm. When the distance between Ag NPs to QDs was less than ~10 nm, SiO_2_@Ag@SiO_2_@QDs showed weaker fluorescence intensities than SiO_2_@QD (without metal) due to the quenching effect. On the other hand, when the distance between Ag NPs to QDs was from 10 nm to 14 nm, the fluorescence intensity of SiO_2_@Ag@SiO_2_@QD was stronger than SiO_2_@QDs due to MEF. These results are expected to be useful for synthesizing multilayer nanocomposites with optimized SERS and MEF effects by the fine control technology of silica shells.

## Data Availability

Not applicable.

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
