# Peer review of "Silica Shell Thickness-Dependent Fluorescence Properties of SiO2@Ag@SiO2@QDs Nanocomposites"

_ijms, 2022, doi:10.3390/ijms231710041_

Round 1

Reviewer 1 Report

The article presents research on the SiO2@Ag@SiO2 nanocomposites with controlled silica shell thicknesses in a wide range and decorated with quantum dots, which I think is interesting and will certainly make thought-provoking reading for researchers of nanocomposites and other materials based on silica nanoparticles.

I think that the manuscript can be published in the IJMS, but it needs minor additions in advance. The following are some issues that I believe should be addressed in the manuscript:

Can the authors provide the mechanism responsible for the different behavior of the two silica precursors tested (TEOS vs. Na2SiO3)?

Why only Na2SiO3 was used in the part of studies where the amount of SiO2@Ag NPs was adjusted and the amount of silica precursor was constant, and vice versa: why for studies with changing amount of silica precursor only TEOS was used? I believe that such an explanation should be included in the paper.

In lines 125-127 the Authors mention that "the thin silica shells were obtained in the range of 4 to 13 nm (...) without leakage of Ag NPs from their surfaces" - I think that it should be explained how was the absence of Ag NPs leakage checked?

Figure 3 - There are two dark blue plots, i.e. for samples QS1 and QS6 - the Authors should consider changing the color for one of the samples – this would improve the readability of the figure.

The sentence in lines 16-18 (Abstract) seems incomprehensible to me – I think there is something wrong with the description of the materials prepared during the study.

Author Response

Dear Reviewer

We appreciate the comments from the reviewer who spents invaluable time and effort. We have incorporated additional modifications based on the reviewers’ thoughtful comments, which has helped us to improve the manuscript. Please see the attachment.

Sincerely Yours,

Bong-Hyun Jun

Associate professor

Department of Bioscience and Biotechnology

Konkuk University, 143-701, Seoul, Republic of Korea

Tel: +82-2-450-0521

FAX: +82-2-3437-1977

E-mail: bjun@konkuk.ac.kr

Reviewer 2 Report

It was a challenging and pleasant activity to review the article "Silica shell thickness dependent fluorescence property of SiO2@Ag@SiO2@QDs nanocomposites" by Eunil Hahm et al.

I would recommend addressing the following comments.

Results and Discussion

_Scheme 1 - Authors must complete the schema legend with the meaning of S1, S3 and S6.

_Figure 1 - I ask the authors' attention to the fact that they must include the information in lines 130-140 in the figure legend. Only then will the reading of this figure make sense; otherwise, the figure is difficult to perceive and conclude there. In figure 1B, you must explain what you mean by "under various conditions". Please reformulate the caption of this figure.

_Line 146 - Why add 70 µg of quantum dots (CdSe@ZnS, QDs) on the surface of SiO2@Ag@SiO2?

_Figure 2 - What is the difference between QS1, QS2, QS3, QS4, QS5 and QS6? Please rephrase the figure caption putting this information.

_Line 178-179- "Quenching is a phenomenon in which Förster resonance energy transfer (FRET) occurs when the distance between the metal NPs and the fluorophore is less than 10 nm." What is the reference of this sentence?

Materials and Methods

_Line 237 - Authors have to complete chapter 3.2.5. Which wavelengths are used? Authors must present all the conditions placed on the equipment to perform the analysis. Please reformulate this subchapter.

_Line 241 - Authors have to complete chapter 3.2.6. Authors must present all the conditions placed on the equipment to perform the analysis. Please reformulate this subchapter.

Graphical abstract

_ The Graphical abstract is not understandable because it is missing the meaning of #1, #2 and #3. Please check.

Author Response

Dear Reviewer

We appreciate the comments from the reviewer who spent invaluable time and effort. We have incorporated additional modifications based on the reviewers’ thoughtful comments, which has helped us to improve the manuscript. Please see the attachment.

Sincerely Yours,

Bong-Hyun Jun

Associate professor

Department of Bioscience and Biotechnology

Konkuk University, 143-701, Seoul, Republic of Korea

Tel: +82-2-450-0521

FAX: +82-2-3437-1977

E-mail: bjun@konkuk.ac.kr

Reviewer 3 Report

This paper describes the Silica shell thickness dependent fluorescence property of SiO2@Ag@SiO2@QDs nanocomposites. In this study, silica shell-coated Ag nanoparticles assembled silica nanoparticles (SiO2@Ag@SiO2) with finely controlled silica shell thickness (4 nm to 40 nm), and quantum dots (QDs) were introduced onto SiO2@Ag@SiO2. The results provide background knowledge for controlling the thickness of silica shells in metal-containing nanocomposites and facilitate the development of potential applications utilizing the optimal plasmonic phenomenon. The results are interesting and suitable for publication. However, the data presentation needs further improvement. I recommend the authors consider to revise the following issues.

1. The fluorescence quantum yield of the QDs should be given. What about the fluorescence properties after the assembly with SiO2@Ag@SiO2 ?

2. Please provide  both the findings of the FTIR and XRD measurements of these nanoparticles (at each step).

Author Response

(The authors gave the same response as above.)

Round 2

Reviewer 2 Report

Very good. 

Reviewer 3 Report

Thank you for adding the information I requested to the article. This manuscript is perfect and completed. I would like to accept it with this form.